# Magneto-Mechano-Electric (MME) Composite Devices for Energy Harvesting and Magnetic Field Sensing Applications

**DOI:** 10.3390/s22155723

**Published:** 2022-07-30

**Authors:** Srinivas Pattipaka, Jaewon Jeong, Hyunsu Choi, Jungho Ryu, Geon-Tae Hwang

**Affiliations:** 1Department of Physics (H&S), Vardhaman College of Engineering, Shamshabad 501218, India; srinivaspattipaka@vardhaman.org; 2Korea Institute of Materials Science (KIMS), Changwon 51508, Korea; jeongjw1204@kims.re.kr; 3Department of Materials Science and Engineering, Pukyong National University, 45, Yongso-ro, Nam-gu, Busan 48513, Korea; sky5021184@pukyong.ac.kr; 4School of Materials Science and Engineering, Yeungnam University, Gyeongsan 38541, Korea

**Keywords:** magneto-mechano-electric conversion, energy harvesting, magnetic field sensor, self-powered devices

## Abstract

Magneto-mechano-electric (MME) composite devices have been used in energy harvesting and magnetic field sensing applications due to their advantages including their high-performance, simple structure, and stable properties. Recently developed MME devices can convert stray magnetic fields into electric signals, thus generating an output power of over 50 mW and detecting ultra-tiny magnetic fields below pT. These inherent outstanding properties of MME devices can enable the development of not only self-powered energy harvesters for internet of thing (IoT) systems but also ultra-sensitive magnetic field sensors for diagnosis of human bio-magnetism or others. This manuscript provides a brief overview of recently reported high-performance MME devices for energy harvesting and magnetic sensing applications.

## 1. Introduction

The multifunctional properties of magnetoelectric (ME) materials could enable the demonstration of novel electronic devices for energy harvesting and magnetic sensing applications. In ME materials, coupling phenomena occur between the magnetic and electric properties, which facilitate manipulation of dielectric polarization by a magnetic field and the control of magnetization by an electric field [1]. Single-phase ME materials are isotropic compounds and chemically homogeneous with an intrinsic ME coupling effect in addition to the co-existence of long-range ordered electric dipoles and magnetic moments [2]. Even though the coupling between polar and magnetic sublattices in the single ME phase would be ideal for the various applications, most single-phase materials have exhibited weak ME coupling performance at room temperature owing to the mutual exclusion of ferroelectricity and ferromagnetism at room temperature [3].

ME composites are composed of physically separated electric and magnetic order materials, which present ME coupling performance with orders of magnitude larger than single phase ME materials at room temperature [4]. The working mechanisms of ME composites have been mainly reported as (i) mechanical strain-based magneto-mechano-electric (MME) conversion, (ii) spin exchange, and (iii) charge carrier [5]. Among these, mechanical strain-driven MME conversion, the focus of this review, has been mostly investigated by many research teams, whereas the studies of the other two working mechanisms are still in the early stages of development.

MME composite devices have been investigated based on combinations of various energy conversion principles such as piezoelectric, triboelectric, and magnetostriction effects [6,7]. When an external magnetic field is applied to an MME device, a mechanical deformation or vibration induced from the magnetostrictive or magnet material is delivered into the piezoelectric or triboelectric material, thus resulting in the generation of electric potential and charges as shown in Figure 1 [8]. The use of MME devices for energy harvesting and magnetic field sensing applications has been actively reported due to their advantages (e.g., energy conversion, output signal, small size, and stable properties) [9,10].

Internet of Things (IoT) sensors will play a key role in the era of the 4th industrial revolution to collect information, analyze this information, and execute an action to enable closely connected automatic systems related to public safety, healthcare, industrial manufacturing, and environmental monitoring [11,12]. Nevertheless, the practical utilization of IoT systems including multifunctional sensors, management circuits, and data loggers in any place is significantly restricted due to the difficulty in securing suitable electric power [13]. For instance, the embedment of capacity-limited batteries in billions of future IoT sensors may be impossible due to the huge labor requirement as well as economic cost to maintain the wireless electronic systems [14]. In this regard, MME conversion devices could be exploited as energy harvesters to actualize self-powered, sustainable, and maintenance-free IoT applications [15]. MME generators based on stray magnetic fields are a promising candidate to demonstrate self-powered IoT systems since MME generators can continuously convert electric energy from alternating current (AC) stray magnetic fields (typically less than 10 Oe) at a fixed frequency of 50 or 60 Hz induced by ubiquitously installed commercial power cables such as electric transmission lines, and cables in factories, buildings, and various other infrastructures [6].

On the other hand, ultra-sensitive magnetic field sensors that enable detection at a sub-pT level have won attention for biomedical magnetic applications such as magneto-cardiography (MCG) and magnetoencephalography (MEG) to perform diagnosis of medical diseases related to the human heart and brain [16]. The living organs generally generate a very tiny AC magnetic field below 10 pT (e.g., 10 pT on a heart, 10 fT on a nervous system, and 100 fT on a brain) with a relatively low frequency range up to 1 kHz [17,18]. The state-of-the-art ultra-sensitive magnetometers that allow for the detection of a sub-pT range are mainly based on superconducting quantum interference devices (SQUIDs) with superconducting loops containing Josephson junctions [19]. However, the SQUID system inevitably requires a liquid-helium coolant for cryogenic superconducting, which increases the volume of SQUID and its production cost, thus significantly restricting the expansion of SQUID for general clinical utilization [20]. Various research teams in the world have investigated MME composite sensors since they present the potential for detecting the pT-range ultra-low magnetic field due to their extraordinary magnetic sensitivity at room temperature with benefits including their simple structure, low cost, and high magnetic directivity [21,22,23].

This paper provides a brief overview of recent progress made in the area of MME composite devices for energy harvesting and magnetic sensing applications reported by our research teams. In particular, high-performance MME generators and sensors based on various advanced materials and device structures such as anisotropic piezoelectric single crystal Pb(Mg_1/3_Nb_2/3_)O_3_-Pb(Zr,Ti)O_3_ (PMN-PZT), textured magnetostrictive Fe-Ga alloy, nanostructured triboelectric polymer films, magnetic field concentrator, hybridized multiple energy conversion structure, etc. are discussed.

## 2. High-Performance MME Composite Devices for Energy Harvesting and Magnetic Field Sensing Applications

### 2.1. Piezoelectric Single-Crystal Crystallographic Orientation for MME Harvesting

The working principle of the conventional MME composite generator composed of piezoelectric and magnetostriction materials is explained in detail. The MME generator placed in a location subject to an external magnetic field induces strain in the magnetostrictive layer by magnetostriction effect (magneto-mechano coupling), and the strain is transferred to the piezoelectric layer which results in electric displacement or dielectric polarization by a direct piezoelectric effect (mechano-electric coupling) [3,24,25]. The sequential interaction process of the MME generator is shown in Figure 2a. The piezoelectric materials used for MME generators in cantilever type structures should have a high transverse piezoelectric effect to achieve higher power density, and this can be achieved by poling (electrically) these materials in the thickness direction due to bending motion resulting from the piezoelectric effect (d_31_ or d_32_ mode). The piezoelectric charge/strain coefficient d_ij_ notation in crystallographic orientation can be understood from the IEEE standard [26].

Single crystal piezoelectric materials show an excellent piezoelectric response due to their uniform dipole alignment. Single crystal Pb(Mg_1/3_Nb_2/3_)O_3_–PbTiO_3_ (PMN–PT) and PMN–PZT are well-known relaxor piezoelectric materials that exhibit a high d_33_ (>2000 pC/N) and a high electromechanical coupling factor k_33_ (>92%) in <001> orientations with a rhombohedral structure near the morphotropic phase boundary (MPB) [27,28,29]. The improved piezoelectric responses of rhombohedral crystals with <001> orientation are attributed to their piezoelectric anisotropy and engineered domain states, which allow for the rotation of <111> polarization toward the <001> direction [27,29]. The enhancement in piezoelectric properties of these single crystals strongly depends on the composition of MPB, crystallographic orientation, and rotation of polarization [30]. These findings suggest that the fabrication of ME composites with particularly oriented single crystals is a good approach for maximizing ME coupling [24,31,32,33].

Based on this concept, Ryu et al. fabricated a high-performance MME generator embedded with PMN-PZT piezoelectric single crystal by employing the piezoelectric d_32_ mode [6]. Various synthesis methods are used for the fabrication of MME generators. The solid state single crystal growth method was used to grow PMN-PZT piezoelectric single crystal with rhombohedral phase. Toxic metal lead (Pb) is hazardous to human health and non-eco-friendly, especially considering this issue with regard to IoT sensor applications in environmental monitoring, biomedical field, etc. [34,35]. These PMN-PZT single crystals are limited in their practical application due to their rigidity and brittleness. To avoid this issue, a high performance flexible piezoelectric single crystal fiber composite (SFC) was fabricated for MME generators as illustrated in Figure 2b. For the fabrication of ME laminates, the SFC was bonded with high purity magnetostrictive Ni plate by thermal cure epoxy resin. The cantilever type structure is considered promising for MME generators at low frequencies due to its simple design, flexibility, the tunability of its resonance frequency, and higher strain transfer. Therefore, the cantilever structure MME generator was designed with piezoelectric SFC, magnetostrictive Ni-plate, and Nd magnetic proof of mass, as shown in Figure 2c,d. The Nd magnets generates magnetic torque in the cantilever structure with clockwise and counterclockwise directions under the AC magnetic field, which could maximize the mechanical vibration of cantilever beam during the MME operation. The magnet as presented in Figure 3 can generate upward magnetic field (B_magnet_), and it could be supposed that a virtual current loop (I_loop_) is flowed with direction of counterclockwise in the mass magnet. When an external AC magnetic field (B_external_) is induced from the right into left, the forces (F) of opposite direction could be applied on the two length sides of mass magnet by Fleming’s left-hand rule. The generated opposites forces on the two length sides of mass magnet make counterclockwise torque motion, which could derive mechanical up bending motion of cantilever structure. Meanwhile, the direction of external AC magnetic field is changed from the left into right, which could make mechanical clockwise torque from the magnet. As a result, the external AC magnetic field enables repeated up and down mechanical bending vibrations of the cantilever beam.

The energy harvesting performance of the very first MME generator was reported in 2015. In this report, the MME with a cantilever structure is tested at resonance frequency in the absence of a direct current (DC) magnetic field. Further, it is tested in the presence of an AC magnetic field (160 μT at a resonance frequency of 60 Hz) generated by Helmholtz coils as shown in Figure 4a. The generated voltages are found to be peak-to-peak voltage (V_pp_) signals of 6.5, 3.4, and 9.5 V for <001>-d_31_, <011>-d_31_, and <011>-d_32_, respectively (Figure 4b). The anisotropic <011> SFC with d_32_ mode has much higher energy generation performance than the <001>-d_31_, and <011>-d_31_ modes. Figure 4c shows the rectified and charged voltage in a 10 μF capacitor as a function of load resistance. The <011> SFC with d_32_ mode exhibits a larger voltage response than other modes. The <011> SFC with d_32_ mode presents nearly three-fold enhancement (−1850 pC/N) relative to <011>-d_31_ SFC (599 pC/N). In addition, a <011>-d_32_ SFC based MME generator with an applied magnetic field of 500 μT at 60Hz turned on 35 high intensity LEDs (Figure 4d). The results of <011>-d_32_ PMN-PZT piezoelectric SFC based MME generator is feasible as a ubiquitous power source for low power portable electric devises (wireless sensor networks and wireless charging systems) by harvesting energy from a living environment.

### 2.2. Effect of Dielectric and Mechanical Loss Factors on MME Harvesting

A variety of figures of merit (FOMs) have been developed to emphasize the selection of piezoelectric materials for energy harvesting applications. The formulas of FOMs of harvesting performance in piezoelectric-based energy harvesters are given by the following equations [6,36].
(1)$$FOM=d_{ij}\times g_{ij}$$
(2)$$FOM_{ij}=\frac{d_{ij}^{2}}{\varepsilon_{ij}^{T}}$$
where *d_ij_* is the piezoelectric charge coefficient, *g_ij_* is the piezoelectric voltage coefficient, and *ε_ij_* is the dielectric permittivity under applied stress T. However, earlier studies reported that the FOMs do not properly explain the output performance of these energy harvesters due to the nonlinear behavior of the piezoelectric materials. Their nonlinear behavior is related to the dielectric and mechanical losses, which result in the degradation of the energy harvester performance [37]. To overcome this issue, many research groups have focused on low-loss piezoelectric single crystals to enhance the energy harvesting performance for MME generators [8,38,39]. MME generators embedded with various piezoelectric single crystals such as high-loss PMN-PZT, medium-loss PMN-PZT, and low-loss PMN-PZT were fabricated and designed [8]. The fabrication methods of these MME generators are similar to the above mentioned in Section 2.1. Figure 5a shows the voltage response as a function of time for various MME generators and tested at an AC magnetic field of 700 μT at 60 Hz. The low-loss PMN-PZT MME generator presents a higher maximum V_pp_ of 94 V than high-loss (V_pp_ = 30 V) and medium-loss (V_pp_ = 42 V) generators. This behavior is attributed to the fact that the magnetoelectric response of the device is dependent on the harvesting voltage as shown in Figure 5b. Therefore, the MME generator embedded with low-loss PMN–PZT has higher harvesting output performance as compared to the other two generators.

Based on this concept, a self-powered electronic circuit system is constructed for charging a mobile phone battery. The system consists of an MME generator, bridge full-wave rectifier, supercapacitor, and DC–DC converter, as illustrated in Figure 5c. The 2.2 mF supercapacitor was fully charged (V_max_ = 20 V and E_max_ = 440 mJ) within 7 min by a low-loss PMN-PZT MME generator, whereas high-loss and medium-loss PMN-PZT MME generators charge the capacitor up to 14 and 17 V (E_max_ = 215 and 318 mJ), respectively, as shown in Figure 5d. The energy harvesting performance of the MME generator greatly improved using a low-loss PMN-PZT SCF due to the reduction of energy losses during energy transformations. In addition, the low-loss piezoelectric material plays a critical key role in producing electricity efficiently. These results confirmed that the self-powered electronic circuit system can store electrical energy from the MME generator and is able to charge a mobile phone battery (Figure 5e), switch on high intensity LEDs, and also run electric fan motor without applying any external source.

### 2.3. The Effect of Texturing of Magnetostrictive Phase O1n the Mme Harvesting Performance

Piezoelectric single crystal and magnetostrictive Ni layer embedded MME generators display electrical output power on the order of a hundred μW under a weak magnetic field [6,8,40,41]. The performance of MME generators is enhanced due to the high anisotropic piezoelectric response (d_32_ mode) of SFC. However, the MME generator output power has been limited by the magnetostrictive Ni layer owing to its relatively low piezomagnetic coefficient. To address this issue, textured magnetostrictive materials can be used in MME composites to improve the harvesting performance. Terfenol-D (chemical formula: Tb_0.3_Dy_0.7_Fe_2_) is a well-known magnetostrictive material that displays a larger magnetostriction of ~2000 ppm under magnetic field of 1 kOe at room temperature [42]. Nevertheless, the presence of a critical rare-earth element (Tb), its brittle nature, and high magnetic bias field for optimal performance have prevented Terfenol-D being used in MME composites. A Fe–Ga alloy with an outstanding magnetostriction property and low production cost displays a larger magnetostriction of ~400 ppm along <100> direction under low magnetic field of ~200 Oe, which is useful for the operation of MME devices under a low magnetic field [43].

The textured magnetostrictive Fe-Ga alloy in MME composites exhibits excellent magnetostriction properties [44]. The Fe-Ga alloy with good ductility and magneto-mechanical coupling was synthesized into thin sheets using a thermo-mechanical processing technique with multi-stage rolling and annealing process (Figure 6(ai)). The fabrication process of the MME generator containing a <100> Fe-Ga alloy sheet and an isotropic Mn-doped piezoelectric PMN-PZT SFC is depicted in Figure 6(aii). The cantilever structured MME generator was designed using Mn-doped PMN-PZT piezoelectric SFC and <100> textured magnetostrictive Fe-Ga sheet with Nd magnetic proof mass. The textured Fe-Ga based MME generator exhibits maximum output V_PP_ of 100 V at the resonance frequency of 100 Hz and α_ME_ of 1330 V/cm·Oe as compared to the conventional Ni-based MME generator (V_PP_ of 59 V and α_ME_ of 781 V/cm·Oe), as shown in Figure 6b,c. The cantilever type MME generators normally have very narrow working frequency range (bandwidth) since it has a fixed resonance frequency which is defined by the materials, dimension of device, mass magnet, etc [45]. The αME value was noticeably decreased into ~1200 V/cm·Oe at ~99.5/~100.5 Hz and ~1000 V/cm·Oe at ~99/~101 Hz. Therefore, the MME composite device can generate the maximum output performance at a specific resonance frequency. Now many research teams have explored wide bandwidth MME composite generators with movable proof mass which could automatically move the position of proof mass to minutely regulate the mechanical resonance frequency of MME generator to match the frequency of the external AC magnetic field. The AC signal generated from the Fe-Ga based MME generator is converted into a DC signal for a demonstration of standalone powered electronic devices. The four diodes are normally utilized to construct the full-wave bridge rectifying circuit. The rectified voltage is saturated (V_DC_ = 76.5 V) at 10 MΩ, whereas the rectified current is decreased from short-circuit current (I_DC_ = 178 μA) with load resistance (Figure 6(di)). A maximum DC output power of 3.86 mW and corresponding power density of 3.22 mW/cm^3^ are obtained at a load resistance of 575 kΩ, as shown in Figure 6(dii). The AC output power of MME composite was 4.6 mW, which was decreased into 3.86 mW after the rectifying process. Therefore, the efficiency of AC-DC rectification was 83.9% in this work. The high output energy generation in the present structure is due to the coupling between strong anisotropic response of piezoelectric SCF and highly textured Fe-Ga magnetostrictive alloy. The energy harvesting response of the textured Fe-Ga based MME generator is much higher compared to the previously reported Ni/low-loss PMN-PZT and Ni/PMN-PZT based MME generators [6,8].

Based on the above concept, a standalone-powered wireless sensor system is constructed, which consists of a textured Fe-Ga embedded MME generator, storage capacitor (2.2 mF), power management circuit, PC monitoring, and sensor module used for data recording and transmission modes as presented in Figure 6e. This MME generator charges the capacitor up to 3 V in 25 s and operates a wireless sensor for data recording and transmission (Figure 6f). The standalone-powered wireless sensor system which can store electric energy from textured Fe-Ga embedded MME generator, is capable of supplying sufficient standalone energy for operating a wireless sensor network system without applying any power source.

### 2.4. Magnetic Flux Concentration Effect on Energy MME Harvesting Performance

The energy harvesting performance of MME composite devises was significantly improved due to the coupling between high anisotropic piezoelectric response of SFC and textured Fe-Ga magnetostrictive alloy. However, the Fe-Ga based MME generator output power has been limited by relatively high permeability of Fe-Ga alloy due induction of stronger eddy currents even at low frequencies, thereby reducing some of the magnetic flux density change, which may reduce power output. Energy harvesting from stray magnetic fields is promising for wireless sensor networks (WSNs) and IoT systems [3,10,41,46,47,48]. It is known that the magnetic flux reaching to the MME generator is very low as compared to the total stray magnetic field due to it spreading in the redial direction around power transmission cables. The performance of MME generators has been continually improved by adopting novel materials, methods, device structures, etc. [7,15,33,49,50,51,52]. A strong magnetic material with higher permeability, known as a magnetic flux concentrator (MFC), can be used to concentrate the magnetic flux on the MME generator. 

In 2020, a MME generator based on MFC concept was reported by Song H. et al. [53]. For the fabrication of MME generator, the piezoelectric SFC (Mn-doped PMN-PZ-PT in <011> orientation with d_32_ mode) was bonded with magnetostrictive Ni layer using epoxy resin. Four pieces of Nd magnets were attached as a proof mass for the optimization point of cantilever structure. After optimization of the geometric structure of the MME generator, the MFC material, design parameter, and 3D location were optimized to examine MFC effect on MME generator using theoretical simulation model. These simulated results confirm that a material with higher permeability (μ_r_ ≥ 600) could be used for the MFC. Hence, Ni with μ_r_ of 600 is chosen as the MFC material [54]. Further, design parameters such as the shape (square: 10 mm × 10 mm), aspect ratio, and number of layers (10 with 2.5 mm thickness) are optimized to obtain a higher response from the MFC (Figure 7a). The output performance of the MME generator is tested with the optimized MFC inside a Helmholtz coil, as shown in Figure 7b. The magnetostrictive layer with MFC (0.691 T, displayed in thick red) presents a 1.5 times larger magnetic flux density distribution than without the MFC (0.465 T, displayed in yellow), as depicted in Figure 7c. The 3D location of the MFC at different distances from the MME generator is optimized by experimental and theoretical simulations to obtain a higher output power response for a standalone powered device. The maximum vibration amplitude of 6.23 mm is found while moving the MFC away from MME cantilever in the upward direction, which is 54% higher compared to the case without the MFC, as shown in Figure 7d. The higher vibration amplitude is attributed to the magnetic flux concentration on MME generator. 

A self-powered wireless environmental monitoring module is constructed and tested under an AC magnetic field of 22 Oe at 60 Hz. It consists of MME generator, MFC, IoT sensor, and power management circuit, and then the system installed at power cable in a substation (Figure 7e). The output energy of the MME generator with the MFC structure presents higher power (5.34 mW) than that without the MFC (2.55 mW), as depicted in Figure 7f. The output power significantly enhanced nearly twice from MME generator by adopting novel method concentrating on magnetic flux around MME generator by tailoring MFC structure including material, shape, aspect ratio, number of layers, and 3D location of MFC. The self-powered wireless environmental monitoring module consisting of MME generator with MFC installed at power station can store electric energy from AC magnetic fields around power transmission cables and it is sufficient to operate a wireless temperature and humidity sensors.

### 2.5. Hybridization of Piezoelectric and Electromagnetic Induction Effects on MME Generator

Even though previously reported piezoelectric crystal-based MME generators presented RMS output power of a few mW level to operate simple IoT devices, this may not be enough to operate high-power consumption (typically scores to hundreds of mW) multifunctional IoT sensors with long range data connectivity that can collect various environmental information such as humidity, light intensity, temperature, air pressure, ultraviolet (UV) index, CO_2_, volatile organic compounds (VOCs), sound level, and magnetic field [55]. Thus, it is significantly desirable to fabricate a high-performance MME generator with an output RMS power above 10 mW to demonstrate standalone powered multifunctional IoT sensor systems. To improve the output energy of energy harvesters, many research teams have explored hybridization of two or three energy conversion mechanisms including piezoelectric, triboelectric, and electromagnetic induction effects [56,57,58]. In this manner, a hybrid-type MME generator is a reasonable solution to noticeably enhance the output power for operation of high-power consumption multi-functional IoT sensors [41].

Figure 8a presents a conceptual illustration of a hybridized MME cantilever generator composed of piezoelectric and electromagnetic induction parts [59]. For the fabrication of the hybrid MME generator, a single crystal PMN-PZT SFC was bonded on a Ti cantilever beam by epoxy adhesive, and NdFeB magnets were used as a magnetic tip mass. The hybridization of the electromagnetic induction part with the piezoelectric part is simply achieved by locating a Cu solenoid coil near the end of the magnets. To maximize the output performance, the second harmonic bending resonance mode is adopted on the hybridized MME generator, which provides the largest displacement of vibration movement at the middle of the Ti cantilever. Figure 8(bi) shows the initial state of the hybrid MME generator before harvesting operation. After the mid-region of the cantilever structure is moved upward with the magnetic-induced clockwise force by the external magnetic field, piezoelectric potential and electromagnetic induction current are generated from the piezoelectric single crystal and Cu coil, respectively (Figure 8(bii)). At the releasing state of the MME cantilever structure (Figure 8(biii)), the mid-region of the Ti cantilever is moved downward with the magnetic-induced anticlockwise force by the external magnetic field, which can derive electric energy generation by tensile stress on the piezoelectric part and rotational movement of magnets near the solenoid coil (Figure 8(biv)). In this work, the output power of the MME generator at the second bending resonance mode is significantly higher compared to that of the first bending resonance mode. To verify the benefit of the second harmonic mode at MME operation, a theoretical COMSOL simulation was conducted, as shown in Figure 8c,d. For the calculation, one end part of the MME cantilever was clamped and a magnet mass was attached to the opposing part in a simplified 2D model. When the MME generator is operated at the conventional first and second bending modes, the resonance frequencies of the first and second modes are 13.3 and 119.5 Hz, respectively, which indicates that the total number of periodic up and down movements of the second bending resonance mode is 10 times higher than that of the first mode during the same time. Figure 8c,d present the simulated movements of the piezoelectric MME cantilever structure and mass magnets at the first and second resonance bending modes, respectively. The maximum RMS output power of the piezoelectric phase at the second bending mode is 2.5 times higher compared to the first bending mode in the simulated results, as shown in Figure 8(ciii). Likewise, the maximum RMS output power of the electromagnetic induction part at the second bending mode is 2.4 times higher compared to the 1st bending mode in the simulated results, as shown in Figure 8(diii).

Figure 8(ei,ii) show the circuit configuration used to measure the RMS output power of the coupled piezoelectric and electromagnetic induction parts with various external load resistors and the real measurement results with the impedance matching condition. As a result, the total RMS output power of the hybridized MME generator is 51.7 mW, which is ~10 times higher than that of previously reported high-performance MME generators. Finally, a standalone-powered IoT environment sensing system is demonstrated by integration of the hybrid MME generator, a power management circuit, and a multi-functional IoT sensor. The remarkable output RMS power from the hybrid MME generator facilitates continuous operation of nine types of environmental sensors and subsequent wireless transmission of the data to a receiver as shown in Figure 8f. This work is the first demonstration of RMS output power from an MME generator exceeding 10 mW.

### 2.6. Magneto-Mechano-Triboelectric Generator (MMTEG)

As mentioned above, the MME generators have been conventionally demonstrated with compositions of piezoelectric materials, magnetostriction materials, and mass magnets. In particular, for the piezoelectric part, single crystal materials such as PMN-PT and PMN-PZT with exceptionally outstanding piezoelectric properties was utilized in the MME generator to achieve high performance. However, the costly and time-consuming production process of a piezoelectric single crystal could be an obstacle to expand the practical utilization of piezoelectric crystal-based MME generators [14]. Energy harvesting materials by utilizing the triboelectrification effect is a promising candidate to substitute piezoelectric materials in MME generators since they can convert mechanical vibration into electric energy with simple, efficient, and inexpensive properties for future self-powered IoT systems [60].

Figure 9a shows the setup image of a magneto-mechano-triboelectric generator (MMTEG) inside the Helmholtz coil to measure the output performance by an induced AC magnetic field [7]. The MMTEG is composed of a PFA (Perfluoroalkoxy alkanes) film, Ti cantilever, Al foil, and proof mass magnets. The Al foil is placed above the PFA attached cantilever structure to derive triboelectrification between the PFA film and Al foil by magnetic-induced periodic mechanical contact vibrations. Figure 9b presents the operation mechanism of MMTEG by adoption of the triboelectric effect and electrostatic induction with the up and down vibrational movements of the cantilever structure under an external AC magnetic field. From the initial state (Figure 9(bi)), the upward vibration of the cantilever structure makes surface contact between the PFA film and Al foil by magnetic force of the magnets under an external AC magnetic field, thus inducing the generation of positive charges on the Al foil and negative charges on the surface of the PFA film, as shown in Figure 9(bii). When the two surfaces are detached by the downward vibration, the opposite charges on the Au electrode layer of the PFA film flow between the two triboelectric parts until a fully released state of the cantilever structure as presented in Figure 9(biii,iv). The upward vibration then reduces the gap distance between the Al foil and PFA film again, resulting in opposite electron flow between the two triboelectric parts. 

To increase the contact charging area to improve triboelectrification in MMTEG operation, a complicated nano-structure is formed on the surface of the PFA film by a salt-nanoparticle aerosol deposition process. To investigate the effect of nano-structures on the MMTEG performance, open-circuit output voltage signals of nano-structured and non-structured PFA films are measured under an external AC magnetic field of 7 Oe at 143.2 Hz. The open-circuit V_pp_ signals of MMTEG devices with and without nano-structure reach up to 708 V and 448 V, as shown in Figure 9c. Figure 9d shows the measured peak power of MMTEGs with and without nano-structures by varying the external load resistances in a range from 1 kΩ to 1 GΩ. The maximum peak power of the nano-structured MMTEG is 21.8 mW, which is 5.7 times higher than that of the non-structured device. To demonstrate a self-powered IoT system, a 1 mF capacitor is charged up to 3.6 V by the output energy of the nano-structured MMTEG within three minutes, and an IoT beacon device is subsequently connected to the capacitor as shown in Figure 9e. The inset of Figure 9e presents the continuous operation of an IoT sensor with a sensing interval of one second by electric energy from the MMTEG.

### 2.7. MMTEG for Optogenetic Neuromodulation

Optogenetic neuromodulation technology has been widely explored globally to remedy neurological diseases including Parkinsonism, somnipathy, and epileptic seizure [61,62]. To demonstrate an ideal optogenetic treatment, flexible microscale light-emitting diodes (f-μLEDs) with chemical/humid/thermal stability, low-heating, and high-power efficiency are considered an outstanding stimulation tool for freely moving animals [63]. Even though an optogenetic system was successfully demonstrated in the form of an implantable device that includes an independent power source of battery, the restricted capacity of the conventional battery could necessitate periodic replacement of the depleted battery every ~five years [64,65]. MMTEGs based on energy conversion of the ambient noise magnetic field into electric energy are a strong candidate to demonstrate self-powered optogenetic biomedical systems [66].

Figure 10a shows a schematic illustration of the MMTEG fabrication process and its optogenetic application by turning on f-μLEDs. A copper oxide (CuO) nanoparticle solution is spin-coated onto a triboelectric Nylon film as a heating amplifier under a flash light photothermal annealing process, which causes thermal surface deformation of the Nylon film to increase the surface area, as shown in Figure 10(ai). Figure 10(aii) shows the morphology of the Nylon surface with a nanoscale bumpy texture and microscale wrinkled structure after etching of CuO particles. Subsequently, a cantilever structured MMTEG is fabricated by utilizing a flash-stamped Nylon film, Ti plate, Teflon film, and NdFeB magnet mass as presented in Figure 10(aiii). By converting an AC magnetic field into electric energy using the MMTEG, implanted f-μLEDs inside a mouse skull can be lit up for optogenetic neuromodulation to induce artificial behavior change (Figure 10(aiv)). To investigate the effect of the flash multiscale structure of the Nylon surface for triboelectric energy conversion, the output performance of flash-stamped and pristine MMTEGs is compared at an AC magnetic field of 7 Oe. The output open-circuit V_pp_ and short-circuit current signals are measured as 870 V/145 μA for the multiscale-structured MMTEG and 638 V/100 μA for the non-structured MMTEG, demonstrating that noticeable performance enhancement of MMTEG is realized after the flash annealing process on the Nylon film. 

Figure 10b presents the experimental setup for powering the f-μLEDs using the flash-enhanced MMTEG under a stray AC magnetic field of 2.1 Oe at 60 Hz induced by an electric wire of a hairdryer. The MMTEG can generate open-circuit voltage and short-circuit current signals of 237 V and 33 μA, respectively, under these conditions, which is enough to continuously turn on the f-μLED device, as shown in Figure 10c. Figure 10d shows a schematic illustration of self-powered optogenetic neuromodulation on a living mouse with implanted f-μLEDs powering by MMTEG. Irradiation of the red light from the LED to the modified cells of primary motor cortex M1, result in neural excitation to induce movement of the mouse whisker. As a result, the self-powered optogenetic neuromodulation by the MMTEG can derive noticeable whisker movement of the mouse, which is tracked by a video capture system and image analysis program as shown in Figure 10(ei,ii). This result indicates that the MMTEG under a tiny AC magnetic field could act as an energy source for biomedical implantable devices by supplementing or replacing conventional batteries.

### 2.8. Ultra-Magnetic Field Sensitive MME Composite Enabled by Flash Photon Annealing

Although sensing of a tiny low-frequency AC magnetic field using MME composites in resonance bending mode is challenging due to their high noise signal level (caused by ambient vibrations), a MME composite can be utilized as an ultrasensitive AC magnetic field sensor with improvement of material properties [67]. To achieve outstanding magnetic detection performance, it is crucial to consider magneto-mechanical conversion properties with magnetic and elastic losses of magnetostrictive materials at a resonance condition [10]. In particular, the magneto-mechanical losses could be reduced by enhancement of the piezomagnetic properties and a mechanical quality factor of magnetostrictive materials via nano-crystallization of phase [22].

Figure 11a illustrates the experimental fabrication process and benefit of surface nano-crystallization of the amorphous FeBSi (Metglas) alloy through a flash photon annealing technique [16]. Note that the conventional thermal annealing technique for nano-crystallization of Metglas to minimize the energy losses normally causes a ductile-to-brittle transition, which would be an obstacle during the device fabrication process as a result of mechanical cracking of Metglas sheets. In contrast, flash photon irradiation on an amorphous Metglas sheet with a suitable high temperature condition and very short (~0.3 ms) pulse duration could produce only a few nano-crystals (size below 10 nm) on the surface of Metglas to enhance the magneto-mechanical properties such as magnetostriction and mechanical quality factor without significant brittleness. Figure 11b presents high-resolution transmission electron microscope (HRTEM) and scanning microscope (SEM, the left top inset) images of the Metglas surface to investigate the effect of flash photon annealing on the microstructure. Due to the partial surface oxidation at the amorphous Metglas surface by flash annealing, dark color zones are partially formed on the SEM image, and nano-crystals with an average crystallite size of 10 nm appeared in the dark color area of the flash photon annealed Metglas. By the nano-crystallization, a few annular-shaped fringes with bright crystallite spots in the fast Fourier transform (FFT) patterns are appeared as shown in the right bottom inset of Figure 11b.

The magnetostriction curves of flash photon annealed and non-annealed Metglas laminations (six-layers) are measured at room temperature, as presented in Figure 11c. The saturated magnetostriction values of the flash photon annealed and pristine Metglas laminates are 30.2 ppm and 36.0 ppm, respectively, which correspond to longitudinal piezomagnetic coefficients of 0.33 ppm and 0.42 ppm, respectively. This enhancement in the piezomagnetic constant of the flash nano-crystallized Metglas is ascribed to (1) the transformation of random alignment into a systematic alignment with the magnetization direction and (2) the elimination of residual stress which usually impedes rotation of magnetic nanodomains. Moreover, Metglas lamination after flash photon annealing shows 26.8% higher mechanical quality factor with a 9% higher hardness value compared to the pristine sample, which is beneficial to obtain outstanding magneto-mechano conversion performance at a resonance condition. Figure 11d shows the device structure of a bilayer MME laminate for an ultra-sensitive magnetic field sensor, which is composed of single crystal PMN-PZT SFC and six-layer Metglas lamination. Finally, the AC magnetic field detection performance of MME composites by flash annealed and pristine Metglas sheets is investigated, as presented in Figure 11e. The flash photon annealed MME sensor displayed a detection limit of 0.5 pT at the resonance frequency of 99.3 Hz, constituting 10-fold enhanced performance compared to the pristine MME sample (5 pT at 97.7 Hz). This outstanding detection limit after flash photon annealing is a combined result of the reduced resonance loss, improved magnetic sensitivity, and resistance to the ambient vibrational noise of the flash-treated Metglas. The flash annealed MME composite is a potential alternative to ultra-sensitive AC magnetic field sensors in the practical application field to detect bio-magnetism and extremely low frequency signals that are hazardous to humans.

## 3. Conclusions

This paper has described several exciting high-performance MME composite devices composed of piezoelectric, magnetostriction, triboelectric, and permanent magnet materials that have been developed for energy harvesters and magnetic field sensors for IoT and bio-medical applications. The MME composites can provide the conversion of stray magnetic fields into an electric signal, and thus can be utilized as high-performance energy generators or ultra-sensitive magnetic sensors. In particular, MME composites with modification of piezoelectric crystals and magnetostriction materials by tailoring the crystallographic orientation or improving loss properties yield a noticeable increment of output power for MME generators. Moreover, the adoption of a magnetic flux concentration structure, and surface-modified triboelectric materials, or hybridization of multiple energy conversion principles in the MME generators enables continuous operation of IoT sensors as well as optogenetic neuromodulation without external energy sources. Even though the MME composite generators cannot operate at ambient radio frequency (RF) electro-magnetic field, they can operate at a relatively low frequency of <200 Hz due to their relatively low mechanical resonance frequency. Furthermore, from the viewpoint of output power, the MME generators can generate electric power of mW-level, which is larger than the previously reported RF energy harvesters (output power of µW-level) [68,69,70]. This technical development could extend the application of MME generators as a permanent electric power supply in self-powered IoT and implantable bio-medical devices to support or substitute conventional batteries. Figure 12 gives a summary on the output power improvement of MME composite generators since 2015. The RMS output power or average AC output power generated from MME generators were used to compared the output performance. The output power of MME composite generators has been gradually improved and recently has achieved up to 50 mW, which is enough to operate multifunctional IoT sensors. However, the output power should be increased above 100 mW for supply of electric energy into environmental gas or chemical IoT sensors to expand the practical utilization field of MME composite generators. The output performance of MME generator may be enhanced by adoption of novel materials including magnetic shape memory alloys which have outstanding mechanical deformation, strain, and response under external magnetic field and mechanical force compared to the conventional magnetostrictive materials [71]. Another technical issue related to MME composite generator is the design of highly efficient power management circuit to convert AC output into DC signal. Researches for advanced electronic circuits including AC-DC rectifier, DC-DC converter, and energy storage unit are therefore significantly required to improve the energy extracted from the MME generators with minimizing energy loss.

As a magnetic field sensor, the MME composite shows very sensitive capability, even permitting detection of a sub-pT range. This sensing performance is enabled by the short-time flash photon annealing process on a magnetostrictive Metglas sheet to induce surface nano-crystallization. This MME-based magnetic sensor demonstrates the potential use of the MME composite sensor as a novel monitoring device for diagnosis of very weak bio-magnetism. In order to directly detect the bio-magnetic field from human heart and brain using the MME composites in the future, it would be necessary to acquire reliable vital data through advanced signal processing methods including data sampling, noise filtering, and signal amplification.

## Figures and Tables

**Figure 1 sensors-22-05723-f001:**
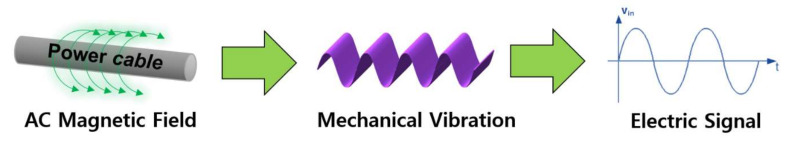
Operation mechanism of magneto-mechano-electric conversion.

**Figure 2 sensors-22-05723-f002:**
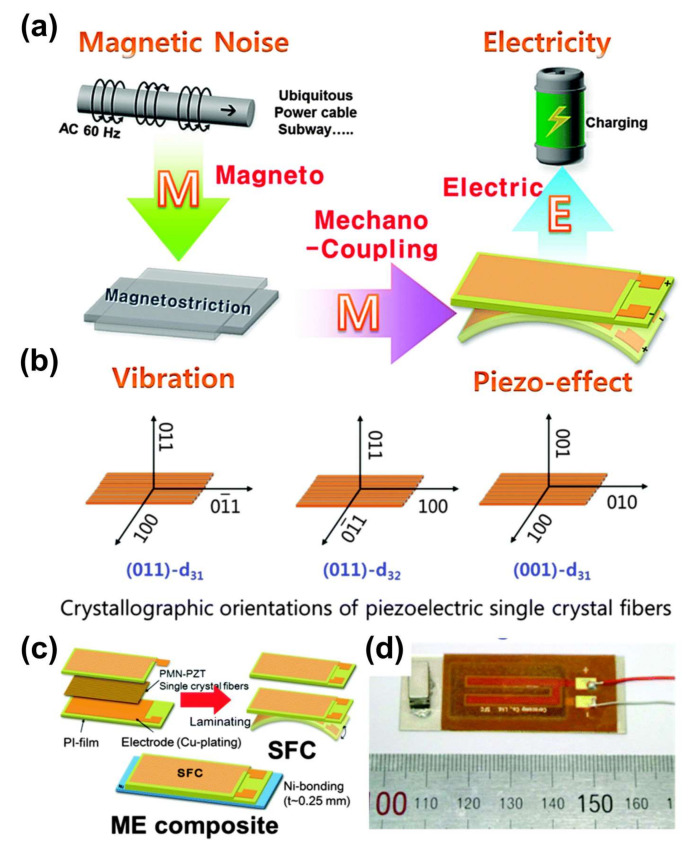
(**a**) Schematic diagram of energy transfer process in MME generator with ME composites. (**b**) Schematic diagram of crystallographic orientations of piezoelectric single crystal fibers used in ME composites. (**c**) Schematic diagram of fabrication steps for the ME composite (**d**) Photocopy of designed MME generator with Nd proof of mass. Reproduced with permission [6].

**Figure 3 sensors-22-05723-f003:**
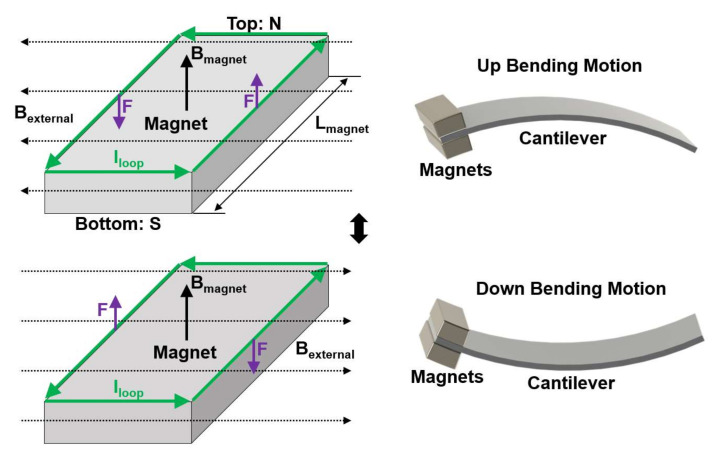
Schematic for mechanism of mechanical vibration on MME cantilever with counterclockwise and clockwise toque on mass magnets under external AC magnetic field.

**Figure 4 sensors-22-05723-f004:**
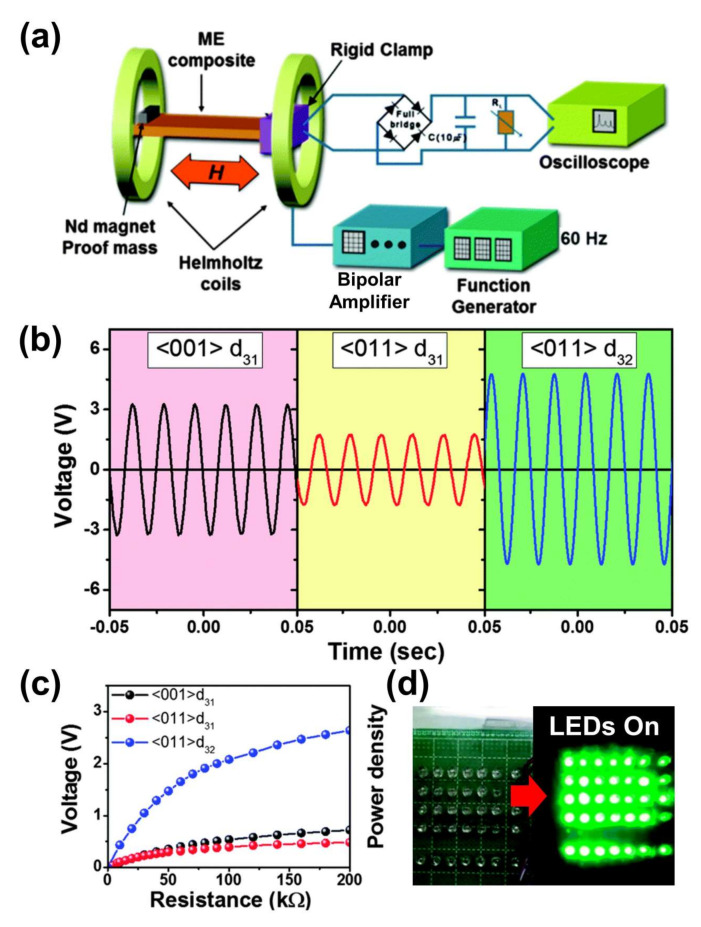
(**a**) Schematic diagram of MME generator measurement setup. (**b**) Output voltage response from MME generator embedded with <011> SFC in d_32_ mode under 160 μT and 60 Hz. (**c**) Rectified voltage as a function of load resistance for various orientations with modes. (**d**) A photograph of 35 LEDs with turn on/off frequency of 1 Hz using the power harvested from MME generator. Reproduced with permission [6].

**Figure 5 sensors-22-05723-f005:**
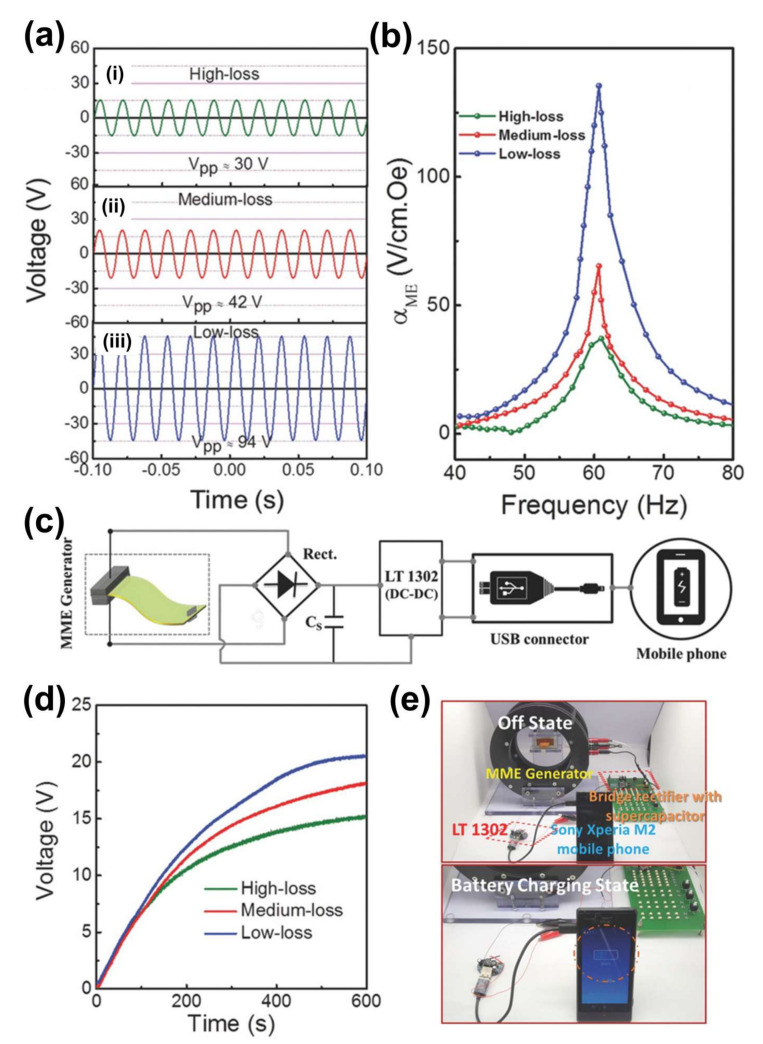
(**a**) Generated voltage from signals from high−loss, medium loss, and low−loss based MME generators. (**b**) ME voltage coefficient (α_ME_) as a function of HAC driving frequency for various MME generators. (**c**) Schematic diagram of self-powered electric system for charging mobile phone battery using MME generator. (**d**) The charging response in 2.2 mF supercapacitor for various generators. (**e**) Off and on state charging phone battery. Reproduced with permission [8].

**Figure 6 sensors-22-05723-f006:**
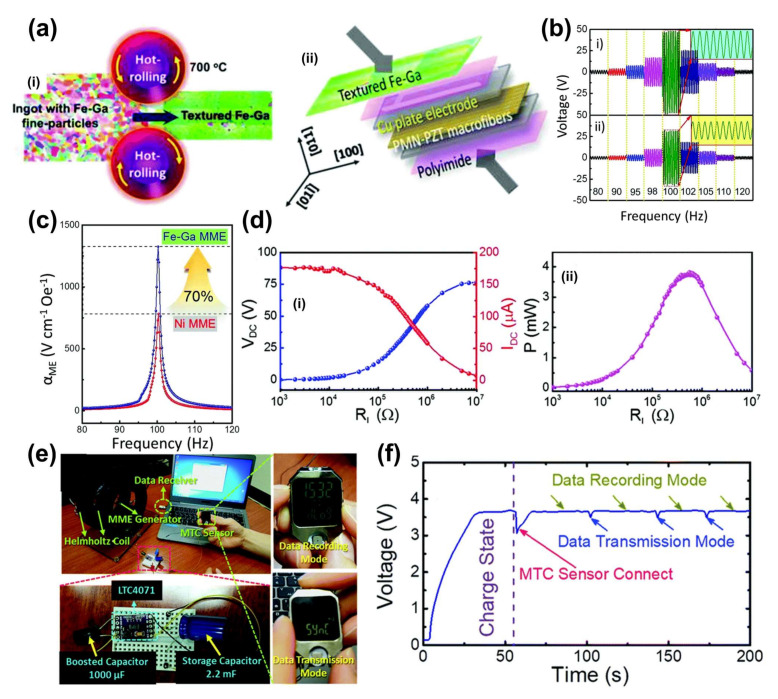
(**a**) (**i**) Schematic diagram of synthesis process of textured Fe−Ga. (**a**) (**ii**) fabrication steps for the ME composite. (**b**) Generated voltage signals from MME generators embedded with textured Fe−Ga (**i**) and Ni (**ii**) layers measured at different frequencies (**c**) ME voltage coefficient (α_ME_) as a function of HAC driving frequency. (**d**) Rectified DC voltage/current signals (**i**) and DC output power (**ii**) of Fe−Ga MME generator under 700 μT at 60 Hz. (**e**) Energy harvesting performance of standalone-powered wireless sensor system using MME generator (**f**) Charging and discharging response of storage capacitor. Reproduced with permission [44].

**Figure 7 sensors-22-05723-f007:**
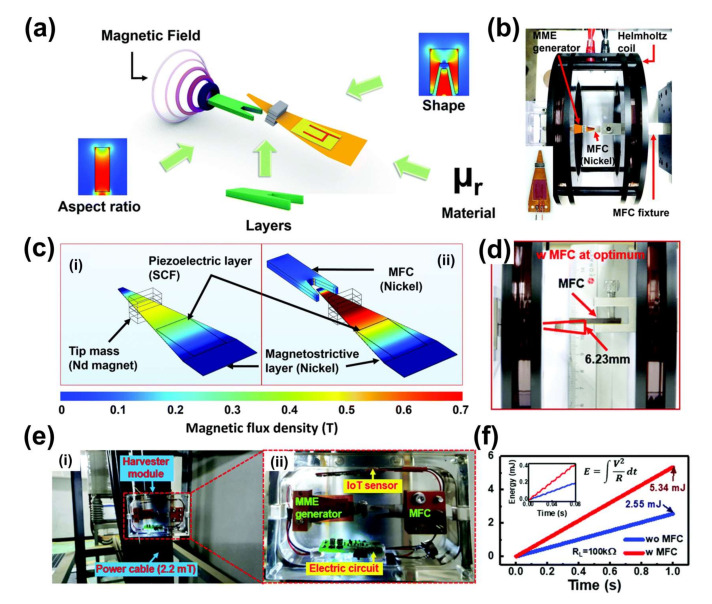
(**a**) Schematic diagram of MFC concentrating stray magnetic field on MME generator with shape, material, layers, and aspect ratio. (**b**) Measurement set-up of MME generator with MFC (**c**) Magnetic flux density distribution in magnetostrictive layer of MME generator with MFC (**i**) and without MFC (**ii**). (**d**) Optimization of MFC location over thickness of MME generator under 8 Oe. (**e**) Photocopy of energy harvesting from a power cable at substation using MFC harvested module. (**f**) Harvested energy response at optimum load resistance of 100 kΩ with and without MFC. Reproduced with permission [53].

**Figure 8 sensors-22-05723-f008:**
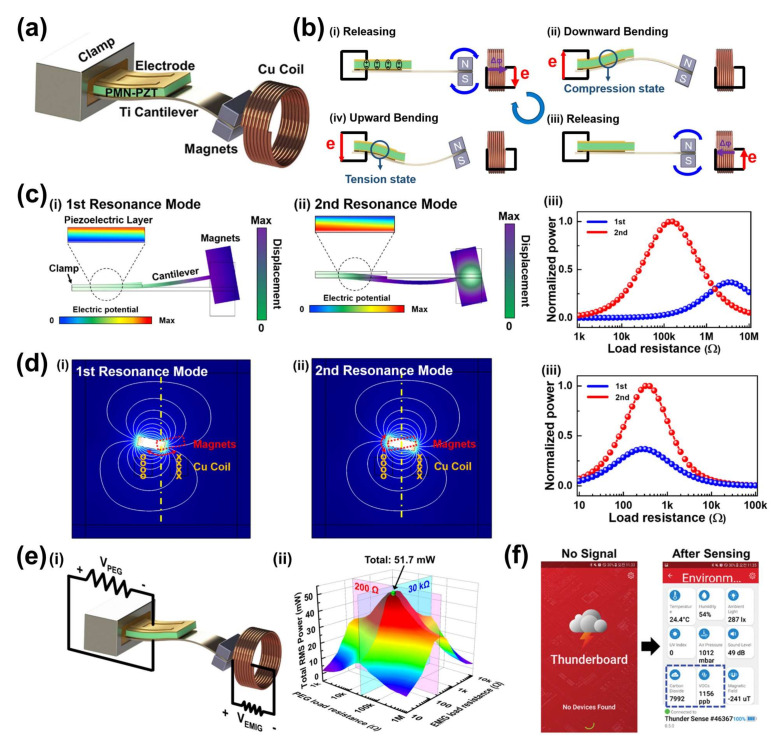
(**a**) Illustration of the hybridized MME generator. (**b**) Schematic illustration of the working mechanism of hybrid MME generator composed of piezoelectric and electromagnetic induction parts. Theoretical MME simulation results of the 1st and 2nd resonance bending modes for piezoelectric part (**c**) and electromagnetic induction part (**d**). (**e**) Circuit configuration for the hybrid MME generator to characterize RMS power with the impedance matching condition (**i**) and total RMS power of the hybrid MME generator with the impedance matching condition (**ii**). (**f**) Snapshot images of IoT monitoring system operated by hybrid MME generator. Reproduced with permission [59].

**Figure 9 sensors-22-05723-f009:**
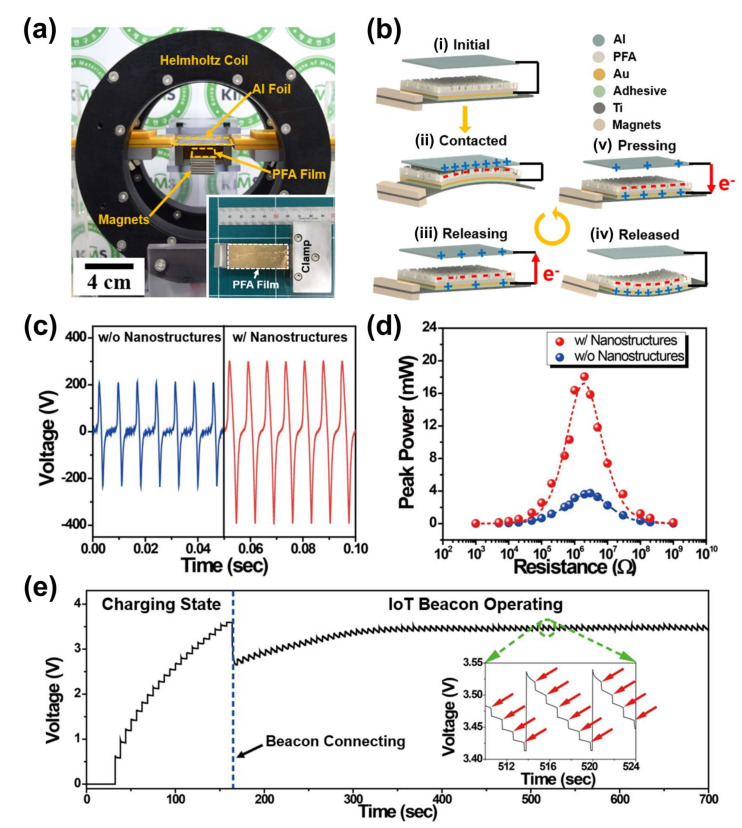
(**a**) Photograph image of MMTEG system installed inside Helmholtz coil. (**b**) Working mechanism of MMTEG under AC magnetic field. (**c**) Open-circuit voltage signals from the nano−structured and non-structured MMTEG devices. (**d**) Peak power values from the nano−structured and non-structured MMTEG devices. (**e**) Charging and discharging curve to operate IoT sensor by MMTEG. Reproduced with permission [7].

**Figure 10 sensors-22-05723-f010:**
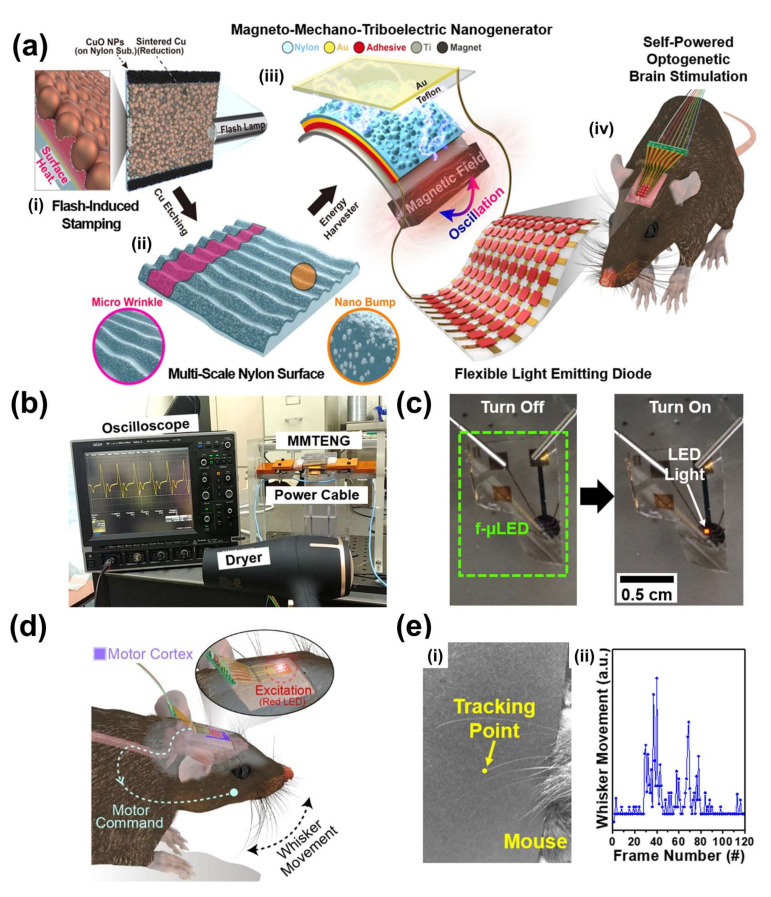
(**a**) Schematic illustration of the fabrication process for MMTEG and its application for optogenetic neuromodulation. (**b**) Photograph image of MMTEG to generate electric energy near power cable of home appliance. (**c**) Photograph of f-µLED turning on by MMTEG. (**d**) Schematic illustration of optogenetic neuromodulation under skull of mouse. (**e**) Video tracking point of optogenetic neuromodulation (**i**) and whisker movement during the procedure (**ii**). Reproduced with permission [66].

**Figure 11 sensors-22-05723-f011:**
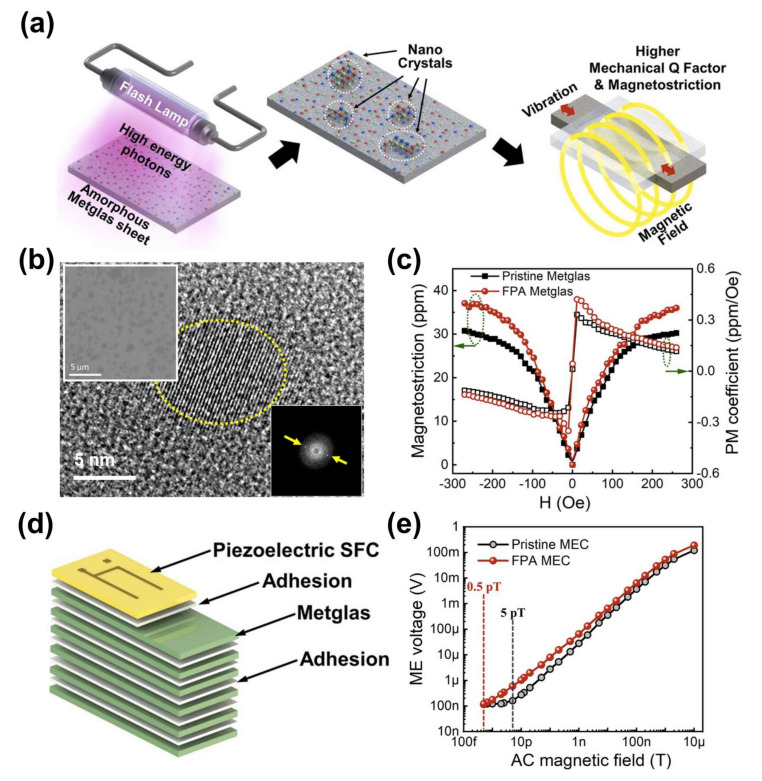
(**a**) Schematic illustration of the flash photon annealing process on Metglas sheet and the expected property enhancement. (**b**) Microstructure analysis results on Metglas surface after flash photon annealing. (**c**) Magnetostriction and piezomagnetic coefficient curves of flash annealed and pristine Metglas laminations. (**d**) Schematic illustration of the Metglas lamination-based MME magnetic field sensor. (**e**) Output voltage response signals of flash annealed and pristine MME sensors by applying AC magnetic field. Reproduced with permission [16].

**Figure 12 sensors-22-05723-f012:**
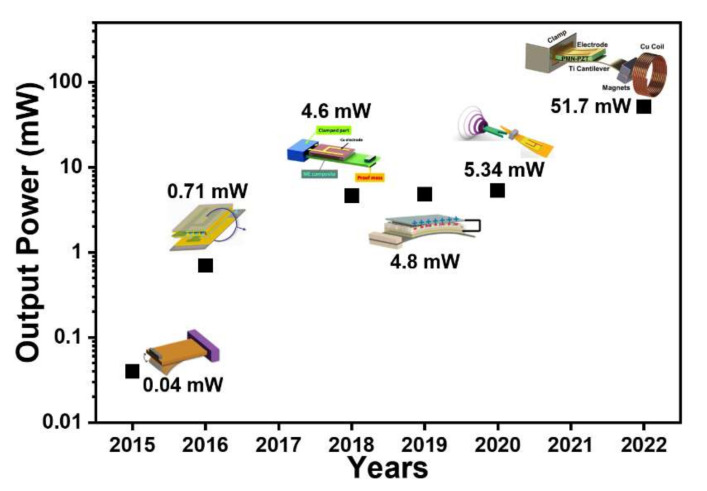
Summary on the output power of MME composite generators since 2015. Copyright 2015, 2018, 2019, 2020, Royal Society of Chemistry. Copyright 2016, 2022, John Wiley & Sons, Inc.

## Data Availability

Not applicable.

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
