# Peer review of "Magneto-Mechano-Electric (MME) Composite Devices for Energy Harvesting and Magnetic Field Sensing Applications"

_sensors, 2022, doi:10.3390/s22155723_

Round 1
Reviewer 1 Report
The authors present an overview of recently reported high-performance MME devices for energy harvesting and magnetic sensing applications. In particular, high-performance MME generators and sensors based on various advanced materials and device structures were discussed. The presentation of the paper in terms of figures clarity and format is very suitable. However, authors have to improve the structure of the paper to be much more clear for the readers. Specifically, a detailed paper structure can be introduced at the end of the introduction.
The presentation of the paper is chaotic, It is difficult to the reader to follow the structure and understand the relation between different sections of the paper. For example, authors presented three main sections as following;
1. Introduction
2. High-performance MME composite devices for energy harvesting and magnetic field sensing applications
3. Conclusion
I think in such case there is no need to define section 2 as a separate section. I recommend to divide it into 2 subsections (so the conclusion will be section 4). For example, authors can make one section for harvesting devices and one for magnetic field sensing applications.
It is also recommended that authors explain at the beginning the methodology of the discussion of the reported results. It is claimed that the selected results are based on the their high performance but authors failed in the discussion of the reported work to present a coherent paper.
Regarding the content of the paper, the first concern of the paper is the definition of “Magneto-Mechanic-Electric” effect and the difference to “Magnetoelectric” effect, if any. The coupling effect between magnetic and electric fields is commonly referred to magnetoelectric effect. Did the authors used “Magneto-Mechanic-Electric” to highlight that they are focusing on this coupling as a composite effect (magnetostrictive + piezoelectric).
In line 27, authors indicated that MME effect is a combination of various conversion principles and they indicated “magnetic force” as one of them. I don’t think that magnetic force is a conversion principle. Please check.
In Figure 1, it is indicated that the AC magnetic field is generated by current through a power cable. AC magnetic fields (mainly for harvesting devices) is created by moving magnets. So please check the figure for more precision.
Other comments:
Figure 2, the abbreviation ME is not introduced in the paper. Please add it.
In figure titles (example fig 4, title of section 2.3 ), please remove “the” in the beginning of the sentences.
Line 163: Figures 4c à Figure 4c
Remove all “.” before references (example in line 217: .[37].à[37].
Author Response
Dear Reviewer, please see the responses in the attachment. Thanks a lot.

Reviewer 2 Report
This manuscript presents a novel magneto-mechano-electric composite devices for energy harvesting. The manuscript is well written with clear organization, novelty, and convincing measurement results. The MME design is also efficient and has enough merits. Specifically, I only have few comments.
1. The authors are suggested to provide more details about the rectifier and its performance.
2. What is the bandwidth of the proposed energy harvester? How broad is the bandwidth?
3. Is the proposed MME energy harvester able to harvest EM energy from higher frequency band, such as RF energy harvesting including 10.1109/JPROC.2014.2357031, 10.1109/LMWC.2015.2451397, 10.1109/TAP.2017.2786320? The authors are suggested to discuss these work and clarify the relationship with RF energy harvesting.
Author Response

(The authors gave the same response as above.)

Reviewer 3 Report
The presented state of the art analysis is relevant but it should be more comprehensive and more valuable by discussing not only strengths, but also emphasizing key weaknesses of different materials, device configurations, etc. The authors should take into account the following comments:
1) The following terminology/descriptions should be corrected: “biomedical healthcare”, PC/N -> pC/N, “Biplar Amp.” in Fig. 3.
2) As drawback, toxicity of Pb should be discussed here “The 108 PMN-PZT single crystals are limited for practical application due to their rigidity and brittleness”, especially considering this issue with regard to mentioned IoT sensor applications in environmental monitoring, biomedical field, etc.
3) As drawback, relatively high permeability of Fe-Ga should be discussed (i.e. induction of stronger eddy currents even at low frequencies, thereby reducing some of the magnetic flux density change, which may reduce power output).
4) The review should be more thorough and include discussion of more novel types of MME generators based on magnetic shape memory alloys (MSMAs), e.g. https://doi.org/10.1016/j.jmmm.2021.168112 .
5) In chapter 2.2 formulas of other FOMs should be given and cited (i.e. FOMs relevant to piezoelectric transducers, vibration energy harvesters).
6) Conclusions should be updated to include short overview of main challenges (as recommendations for future research) and other issues/limitations that should be addressed by scientific community in order to further advance MME generator and sensor technology.
Author Response

(The authors gave the same response as above.)

Round 2
Reviewer 1 Report
The authors have satisfactorily addressed most of my concerns. This revision has significantly improved the manuscript.